# Estimation and Potential Analysis of Land Population Carrying Capacity in Shanghai Metropolis

**DOI:** 10.3390/ijerph19148240

**Published:** 2022-07-06

**Authors:** Hefeng Wang, Yuan Cao, Xiaohu Wu, Ao Zhao, Yi Xie

**Affiliations:** School of Mining and Geomatics Engineering, Hebei University of Engineering, Handan 056038, China; wanghefeng@hebeu.edu.cn (H.W.); wuxiaohu1027@163.com (X.W.); 18171895455@163.com (A.Z.); xieyihue@126.com (Y.X.)

**Keywords:** urban land carrying capacity, population estimation, land use space, potential analysis, Shanghai Metropolis

## Abstract

It is of great practical significance to understand the current situation of urban land carrying capacity, explore its potential space, and continuously improve the economic adaptability and resilience and population carrying capacity of megacities. Based on the guiding principle of territorial spatial division and the concept of moderate-scale resilient cities, combined with GIS technology, this study aims to divide land spaces into three types and construct different index systems to evaluate the land carrying capacity of Shanghai in different spaces. Furthermore, we propose different schemes of estimating subspace land population carrying capacity, and the carrying potential of land population is analysed as well. The acquired results demonstrate three key points. Firstly, the total land population capacity of Shanghai is estimated at 25,476.61–32,047.27 people, with urban land space being the most dominant for the city’s population carrying capacity. Furthermore, the inner suburbs carry the largest population, and the urban centre carries a larger population density than other areas. Secondly, there are significant spatial differences in land population carrying potential. Compared with the demographic data from 2017, Shanghai still has a population carrying potential of 1293.30–7863.97 people and a suitable population carrying potential of 4578.64 people. The population of the urban centre is near the upper limit of the estimated population carrying capacity, and the suburbs, especially the outer suburbs, have large population carrying potential. Thirdly, the estimation method adopted in this study can effectively reveal the spatial differences in population carrying capacity and the potential of different land spaces and different regions in Shanghai, with the estimation results being highly credible. The results will provide references for the improvement of the multi-scenario population planning strategy in Shanghai, as well as enrich the research span and methods currently employed in land carrying capacity.

## 1. Introduction

Arrow et al. systematically discussed land carrying capacity in “Economic Growth, Carrying Capacity, and the Environment,” published in *Science* with a far-reaching impact [1]. Land carrying capacity is defined as the limits of the number of people and the scale and intensity of human activities that land resources can carry under given economic, social, technological, and ecological environment conditions [2]. Due to rapid urbanization, as well as resource and environmental constraints, the carrying capacity of urban land remains unclear. Scholars have conducted an abundance of relevant research, predominantly reflected in the following four aspects. Firstly, studies have dealt with carrying capacity development. More specifically, studies shifted from biological population carrying capacity to land carrying capacity marked by grain, that is “arable land–food–population,” carrying capacity research [3,4,5,6]. With urbanization and industrialization, land carrying capacity has shifted away from the upper limits of population that can be carried by single pieces of cultivated land and moved closer to the comprehensive socio-economic activities that land can support [7,8]. Secondly, relevant studies have focused on evaluation factors and contents of land carrying capacity. The evaluation factors are selected mainly from the aspects of environment, land, water, transportation, social economy, and infrastructure, while the core single-factor evaluations shift to the comprehensive evaluation of the multi-factor index system. Several studies have also considered urban capacity, scale constraints, and other related factors that are primarily used to estimate the urban land carrying capacity index. In addition, land carrying capacity evaluation factors and contents are used to analyse the economic or population scale that may be carried under urban sustainable development. However, there have been relatively few studies conducted on the quantitative estimation of land carrying economy or population scale [9,10,11]. The third relevant aspect is the study scales of land carrying capacity; the aforementioned envelops scales of counties, cities, provinces, urban agglomerations, countries, and even the world [10,12,13,14]. Lastly, the fourth relevant aspect concerns the carrying capacity evaluation methods. Presently, there are various evaluation methods and models of urban carrying capacity in three categories. The first of these is the index system method, which forms a hierarchical structure by combining a series of indicators reflecting all aspects and interactions of the urban carrying capacity. It obtains the comprehensive carrying capacity evaluation index by weighted summation layer by layer [15,16,17,18]. The second category is the model method, which uses the index system, algorithm, and weight combination to simulate and abstract urban carrying capacity. Commonly used methods within the second category include the conventional trend method, fuzzy mathematics, the artificial neural network, the grey prediction model, and system dynamics [19,20,21,22,23]. The third category refers to the spatial analysis method, which uses GIS and RS technology to process the spatio-temporal data of various factors, comprehensively analysing spatial differences in urban carrying capacity [24,25]. In addition, some scholars have carried out a series of studies on land population carrying capacity. For example, Feng and Cheng et al. have estimated China’s land population carrying capacity at different times and space scales [26,27]. They have also analysed possible influencing factors based on arable land and human–food relationships. Fan et al. used different calculation models to estimate the appropriate population of Xi’an based on the comprehensive land carrying capacity [28]. Fu and Dong predicted the land population carrying capacity of districts and counties in Hexi Corridor from 2015 to 2024 using the grey prediction model modified by background value [29]. All the previously mentioned studies calculated and predicted land population carrying capacity based primarily on the perspective of cultivated land and grain.

Land population carrying capacity is a comprehensive index, revealing the relationships between people and land, and the mutual restrictions and promotions between urban population and land are significant themes in geography. The existing research on land carrying capacity has made clear progress. However, there are still many issues in need of further study with respect to the evaluation ideas, estimation methods, and predictions of urban land carrying capacity. Firstly, with the development of the social economy and the development intensity of land use space, land carrying capacity shows dynamic changes with the time node that have not been well understood. Secondly, the present research regards land carrying objects (economy or population) as a fixed value rather than an interval range, which is unreasonable to some extent. Thirdly, few studies construct the evaluation index system of carrying capacity with its own characteristics and quantitatively estimate land carrying economies or population scales based on the land use function and urban natural endowments. Fourthly, the quantitative analysis of the spatial differentiation of land carrying capacity and the potential help of GIS technology have not been thoroughly developed.

At the same time, we also note that United Nations Sustainable Development Goal 11 mainly focuses on cities and puts forward the goal of the safety, resilience and sustainability of cities and human settlements to tackle the challenges of environmental and other disasters. However, with rapid urbanization and urban agglomeration, the scale of cities has expanded drastically, with central cities and urban agglomerations becoming the main spatial forms of development. The population pressure of urban agglomerations and central cities is excessive, with “urban diseases” such as land contradiction, air pollution, traffic congestion, water shortage, and public safety becoming more serious and the comprehensive carrying capacity of cities becoming concerning. As the central city of the Yangtze River Delta, Shanghai megalopolis, is in need of the Shanghai Urban Master Plan (2017–2035) to control the scale of resident population and limit the population to about 25 million by 2035. The plan proposes exploring and improving the multi-scenario planning strategy in order to regulate the matching relationship between population and land scale to cope with the uncertainty of future economic development and population changes. Therefore, extensive research should been done with respect to the questions of whether Shanghai’s actual population scale is appropriate to the population scale that the city can carry, as well as how large the preset population growth space of the city needs. To this end, Shanghai is selected as the research object of this study, combined with the Shanghai Urban Master Plan (2017–2035), and 2017 as the base year. The paper studies the evaluation of subspace land carrying capacity, population estimation, and potential analysis under the constraints of natural resources and socio-economic development, in order to uncover the differences of land population carrying capacity and the potential of different land spaces and regions in Shanghai. Simultaneously, it provides a method of reference for other relevant research and contributes to the overall study of the relationships between humans and land.

## 2. Materials and Methods

### 2.1. Data Source and Data Processing

In accordance with the selected evaluation indicators, the data mainly includes statistical data and spatial data, as shown in Table 1. Based on the vector data of the Shanghai administrative divisions, the geo-referencing, coordinate transformation, and digitalization are processed to obtain spatial data. Shanghai comprehensive evaluation map of surface soil environmental quality, Shanghai suitability zoning map of natural foundation engineering construction, Shanghai potential geological disaster zoning map, contour map of Shanghai land subsidence, and the comprehensive evaluation map of water quality districts of Shanghai in 2017 need to be digitized. Different values are assigned to the corresponding vector layer attribute tables in accordance with the way that the higher levels in the original data are assigned higher values. The road network density data is obtained by digitizing traffic maps and statistically calculating the quantity of road miles per unit area of administrative units.

In addition, combining with the definition of wetlands provided by Kirkman et al. and Li [30,31], the land use types of wetlands (paddy field, river surface, lake surface, reservoir surface, coastal mudflat, inland mudflat, marshland) are extracted from the Shanghai Land use status data for 2017, while the proportion of wetland areas can be obtained by statistically calculating the ratio of wetland area compared to the total area of administrative units. In addition, the industrial land and cultivated land areas are extracted by using similar methods, with the proportion of industrial land and per capita arable land area also being calculated. The concentration of cultivated land is calculated by quoting the calculation formula of the geographical concentration index [32,33]. The formula is as follows.
(1)$$G=100\times \sqrt{\sum_{i=1}^{n}{\left(\frac{x_{i}}{T}\right)}^{2}}$$

Above, G is taken to be the regional cultivated land concentration index, while *n* is the quantity of cultivated land plots in the region and $x_{i}$ is the cultivated land area of plot *i*. Lastly, *T* represent the total area of cultivated land in the region.

The downloaded DEM data and landsat8_OLI image data were processed by image mosaicing, coordinate transforming, resampling and clipping. The slope data for Shanghai was obtained by slope calculation in the GIS software. The normalized differential vegetation index (NDVI) of Shanghai was calculated by using Equation (2) with remote sensing software [34]. Based on the this, the vegetation coverage of Shanghai was calculated by Equation (3) [35].
(2)$$\mathrm{NDVI}=\frac{b_{\mathrm{NIR}}-b_{R}}{b_{\mathrm{NIR}}+b_{R}}$$
(3)$$f_{c}=\frac{\mathrm{NDVI}-\mathrm{NDV}I_{s}}{\mathrm{NDV}I_{v}-\mathrm{NDV}I_{s}}$$

In the above equation, $b_{R}$ represent the reflectance in red band, while $b_{\mathrm{NIR}}$ represents the reflectance in the near infrared band. Next, $f_{c}$ is taken to be the vegetation coverage (%) and $\mathrm{NDV}I_{v}$ is the NDVI value of total vegetation coverage, while $\mathrm{NDV}I_{s}$ is the NDVI value of bare soil. In most cases, the NDVI value of vegetation coverage area is above 0.7, while the NDVI value of bare ground is around 0.03–0.06. In the paper, the $\mathrm{NDV}I_{v}$ = 0.7 and $\mathrm{NDV}I_{s}$ = 0.05 for calculation. If $\mathrm{NDV}I_{v}$ > 0.7, it is assumed that full vegetation coverage $f_{c}$ = 1, and if $\mathrm{NDV}I_{v}$ < 0.05, it is assumed that bare ground $f_{c}$ = 0 [36].

Taking the district as the unit, the statistical index data is imported into the attribute table of Shanghai district boundary layer, and the basic database of statistical indexes is established. All evaluation index data is rasterized, and the raster unit size is 30 m × 30 m. Furthermore, the raster layer of each index is obtained, and maximum difference dormalization method is employed to normalize the raster layers, in order to prepare data for subspace evaluation.

### 2.2. Methods

#### 2.2.1. Division of Land Use Space

In 2015, the “Technical specifications and preparation guidelines of the master plan for economic and social development of cities and counties (for Trial Implementation)”, jointly issued by the State Bureau of Surveying and Mapping and National Development and Reform Commission of the People’s Republic of China, emphasizes the need to strictly implement the main function of cities and counties. Moreover, by combining the administrative and natural boundaries, the plan divides the three types of land spaces (urban, agriculture, and ecology) for cities and counties by using the standard and unified geospatial data and the results of a general investigation on geography and national status. The above guideline clearly proposes dividing the territorial space into three types: urban space, agricultural space, and ecological space for the first time, and the corresponding land use types of different land use spaces are given. The Shanghai Urban Master Plan (2017–2035) also optimizes ecological, agricultural, and urban spaces and implements the strategy of main functional areas. In addition, some scholars are actively exploring the land classification standards for territorial spatial planning and thus building the framework of territorial spatial classification from the top-level design. They propose adjusting the original three categories of “agricultural land, construction land, and unused land” to “agricultural land, construction land, and ecological land”, and optimizing the original 12 Level I types and 57 Level II types (Table 2) [37,38].

According to the main ideas and concepts of territorial spatial division stated above, this article divides Shanghai’s territorial space into urban land space, agricultural land space, and ecological land space. Combined with the “Current Land Use Classification Standard” (GB/T21010-2017) [39], land spaces are divided according to the land use codes of level I types. The specific land type merging scheme is provided in Table 3. There are no four land types of natural pasture, glacier and permanent snow, saline, and marshland in the Level II types of Shanghai Land use status. Although parks, green areas, and scenic facility land may be used as land for ecosystem service functions, they are predominantly used for public service functions. Hydraulic construction land is the Level II types of water area and water conservancy facilities land. This type of land mainly refers to the building land above the shoreline of the normal water level such as artificially built gates, dams, embankment forests, hydropower plants, and water lifting stations, all classified as urban land space in the paper. The Level II types of other land are assigned to the corresponding land spaces according to land use function. Based on Shanghai Land use status in 2017 and with the deduction of the river surface area, Shanghai Land use status is combined into urban land space, agricultural land space, and ecological land space by using GIS software (Figure 1). The total area of the three types of land spaces is 7071.693 km^2^, while the area of each land space is shown in Table 3.

#### 2.2.2. Evaluation Indicator Selection

In a sense, the evaluation of different land spaces is the heterogeneity analysis of urban land use functions, and the evaluation indicator system should consider the functions and utilization intensity of each land space, as well as reflect the background support and constraints. Therefore, combined with background elements characteristics of land resource, water resource, ecological environment, geological environment, and the current problems in Shanghai, and the subspace evaluation indicator systems of land carrying capacity are established from the functions of the three types of land spaces, respectively (Table 4). The selection basis for each land space evaluation indicator system is as follows:

Taking into consideration the socio-economic development level, land development intensity and constraints in the urban land space, the following ten indicators are selected to build the evaluation indicator system. Among them, per capita construction land and industrial land proportion reflect urban construction scale and land structure, while population density, economic density and urbanization rate embody the level of social economy and urbanization. In addition, hospital beds per 10,000 people, road network density, and infrastructure investment per unit area illustrate the infrastructure investment of urban land space, while the suitability of natural foundation engineering construction and land subsidence imply the constrains of urban construction.

In agricultural land space, the establishment of the evaluation indicator system reflects the guaranteed degree of cultivated land and food requirements, output efficiency, and natural constraints on agricultural production. The cultivated land concentration indicator is selected to reflect the spatial distribution pattern of cultivated land, while the per capita cultivated land implies the per capita occupancy of cultivated land. Furthermore, the per capita food occupancy reflects the guaranteed degree of food requirements. Lastly, the agricultural labor productivity and output value per unit of agricultural land illustrate the agricultural output level, while the soil environmental quality and slope represent the natural constraints on agricultural production.

In ecological land space, indicators are selected to reflect the ecological resource endowment, ecological sensitivity and pressure status. The air quality index and PM2.5 are selected to show the atmospheric environmental quality, while the comprehensive water quality index reflects the hydrological environment. Moreover, the per capita park green area, vegetation coverage, and wetland area ratio mainly reflect the ecological resource endowment. Finally, the geological hazards susceptibility represents the spatial distribution characteristics of a potential geological disaster in Shanghai.

#### 2.2.3. Evaluation Indicator Weighting

To avoid the influence of subjective factors, this paper employs the entropy method to calculate the indicator weights. This method represents an objective weight determination method, as it determines the weight of indicators according to the amount of information provided by the observed values. When there is a major difference between the indicator values of different objects, the entropy value is smaller. This shows that the indicator provides more effective information and its weight should be larger, and vice versa. Furthermore, when the indicator value of each object is exactly the same, the entropy value reaches the maximum. This means that the indicator has no available information, and can thus be removed from the set of evaluation indicators [40]. The weighting steps of the entropy method are as follows:

First, set the original data matrix as $A={\left(a_{ij}\right)}_{m\times n}$, in which m represent the number of evaluation objects and n is the number of evaluation indicators. The matrix is normalized to $R={\left(r_{ij}\right)}_{m\times n}$.

If the indicator is positive, the formula is shown below:(4)$$r_{ij}=\frac{a_{ij}-\underset{j}{\min}\{a_{ij}\}}{\underset{j}{\max}\{a_{ij}\}-\underset{j}{\min}\{a_{ij}\}}$$

If the indicator is negative, the formula is shown below:(5)$$r_{ij}=\frac{\underset{j}{\max}\{a_{ij}\}-a_{ij}}{\underset{j}{\max}\{a_{ij}\}-\underset{j}{\min}\{a_{ij}\}}$$

The matrix $R={\left(r_{ij}\right)}_{m\times n}$ is normalized by column vectors to obtain $F={\left(f_{ij}\right)}_{m\times n}$.

2.The entropy value of indicator is shown below:


(6)
$$e_{j}=-\frac{1}{\ln m}\sum_{i=1}^{m}f_{ij}\ln(f_{ij})$$


3.The difference coefficient of indicator is shown below:


(7)
$$g_{j}=1-e_{j}$$


4.The weight of indicator *j* is as follows:


(8)
$$w_{j}=\frac{g_{j}}{\sum_{j=1}^{n}g_{j}}$$


Based on the above steps, the average value of each index of the administrative units is used in the weight calculation, and the evaluation indicator weights of the three types of land spaces in Shanghai are obtained. This is shown in Table 4.

#### 2.2.4. Estimation Method of Subspace Land Population Carrying Capacity Based on GIS

By combining the existing research with the guiding principles of territorial spatial planning, the space division of land use in 2017 is carried out according to the land functional attributes. The evaluation indicator systems of land carrying capacity are constructed for different land spaces, while the aforementioned entropy method is employed to determine the weights of different land space indicator systems. Furthermore, on the basis of rasterizing and standardizing the spatial data of each evaluation indicator, the state indexes of subspace land carrying capacity are calculated by using GIS software. As a result of the above mentioned, the evaluation and grading results of subspace land carrying capacity are obtained. On this basis, the estimation schemes of the per unit area population carrying threshold are established for different land spaces and different grades. Lastly, the population carrying capacity interval of each land space is estimated, while the land population carrying capacity of different regions is statistically analyzed. The estimation method flow is illustrated in Figure 2.

## 3. Results

### 3.1. Analysis of Estimation Results

#### 3.1.1. Evaluation Results of Subspaces Land Carrying Capacity

By using the GIS spatial analysis tool, the vector data for each land subspace is masked with the corresponding evaluation indicator raster data in order to obtain the raster data for each land subspace evaluation indicator. The raster calculator is used in combination with the weight of each indicator to obtain the carrying capacity state indexes of the three types of land spaces. The evaluation results are provided in Figure 3.

With regard to urban land space (Figure 3a), land carrying capacity shows a decreasing trend from the urban centre across the inner suburbs to the outer suburbs. The high state index area of carrying capacity is primarily concentrated in the urban centre, among which Jing’an and Huangpu districts are the highest. Secondly, the status index of carrying capacity in the outer suburbs is generally low, with the status indexes of carrying capacity in Qingpu, Chongming, Songjiang and Jinshan Districts being less than 0.156, which is the lowest region of land carrying capacity in urban land space. The carrying capacity state index of Minhang district in the inner suburbs is higher than that of Baoshan, Jiading and Pudong New districts. Furthermore, although Lujiazui in Pudong new district has a high level of economic development and better infrastructure construction, the state index of carrying capacity is also relatively low due to the local and overall natural resource endowment of Pudong new district.

In agricultural land space (Figure 3b), the carrying capacity state indexes of the Jinshan and Chongming districts are higher those of other districts in the inner suburbs, followed by the Qingpu, Fengxian, and Songjiang districts. Pudong New district (mainly, the original Nanhui District) exhibits relatively high land carrying capacity, while the carrying capacity state indexes in the Jiading and Baoshan districts are lower. Spatially, the high carrying capacity of agricultural land space is scattered across different directions in Shanghai. On the whole, the land carrying capacity in the outer suburbs is higher than that in the inner suburbs.

In ecological land space (Figure 3c), the carrying capacity state indexes in Chongming district, which has a good ecological environment is the highest, followed by the southwest of Qingpu district, Fengxian district, the east of Jinshan district, Pudong new district, and Huangsha Island. Conversely, the state index of carrying capacity in the urban centre is lower. Furthermore, index exhibits a descending trend from the outside in, indicating that the ecological environment is better the further away from the urban centre it is. The index also indicates that the ecological environment in the urban centre needs further improvement.

#### 3.1.2. Grading Evaluation Results

In order to distinguish the carrying capacity spatial differences of each land space further, the evaluation results of the three types of land spaces are assigned grades, and the grade number is determined as five. In order to make the differences between grades more evident, the grading method has adopted the natural breaks (Jenks) method, which grades the data most appropriately based on its own characteristics. According to the carrying capacity evaluation results of the three types of land spaces, the break points are detected in ArcMap to determine the interval values. According to the grade interval value of each land space, the reclassification module of the spatial analysis tool is used to reclassify the carrying capacity evaluation results of the three types of land spaces. The classification results are shown in Figure 4, while the interval values, grade names and statistical results of each grade area are shown in Table 5.

In urban land space, the areas of “Good” and “Better” grades of land carrying capacity are small at values of 6.42% and 2.62%, respectively. And the areas of “Inferior” and “Poor” grades combine to account for 73.28%. According to Table 6, the urban centre only includes “Good” and “Better” grades, with their areas accounting for 99.50% of the total area of the two grades. In addition, there is only 1.385 km^2^ of the “good” grade in the inner suburbs. The grade in the inner suburbs is mainly “Inferior”, accounting for 80.58% of the total area of this grade, followed by “Common”. There is no “Better” grade distribution, while a small amount of “Good” and “Poor” grades distribution. There are only two grades of “Inferior” and “Poor” in the outer suburbs, and most of the “Poor” grade is distributed in the outer suburbs, accounting for 99.7% of the total area of this grade.

In agricultural land space, the areas of “Better” and “Common” grades of land carrying capacity are the largest, accounting for 44.94% and 38.96% of the total area of land space, respectively. The two are followed by the “Good” and “Poor” grades, while the area of the “Inferior” grade of land carrying capacity is the smallest, accounting for only 1.45% of the total area of land space. Furthermore, the land carrying capacities of Chongming and Jinshan districts are shown to be the best, with the “Better” grade almost entirely distributed in these two districts. The “Good” grade is mainly distributed in Qingpu district, while the “Poor” and “Inferior” grades are found in the Jiading and Baoshan districts, which have weaker land carrying capacity.

In ecological land space, the area of the “Poor” grade of land carrying capacity is the smallest, accounting for only 1.04% of the total area of land space. Furthermore, the areas of “Good” and “Better” grades are the largest, accounting for 40.56% and 34.2% of the total area of land space, respectively. The above two grades are followed by the “Inferior” and “Common” grades. While the “Better” grade is mainly distributed on Chongming Island, the “Good” grade is concentrated in the Pudong New district, Fengxian district, the east of Jinshan District, and the southwest of Qingpu District. Lastly, the “Poor” and “Inferior” grades are located in the urban centre.

### 3.2. Estimation Results of Land Population Carrying Capacity

#### 3.2.1. Basic Concepts of Estimation

Combining the population density of domestic and foreign metropolises and Shanghai, the research results of relevant scholars and the requirements of the Shanghai Urban Master Plan (2017–2035). Firstly, the benchmark interval value of the population carrying capacity per unit area is determined for the “Common” grade of land carrying capacity evaluation. Furthermore, the population carrying standard per unit area for other grades is adjusted on the basis of the above mentioned. While determining the grade carrying standard, the internal differences of the land carrying capacity evaluation grades in different land spaces and the positioning and advantages of different location functions should be considered, and the carrying standard should be appropriately adjusted. Finally, according to the statistical areas of different grades, the population carrying capacity interval of each land space is calculated respectively, and the total land population carrying capacity interval of Shanghai is summarized.

#### 3.2.2. Estimation Scheme

##### Estimation Scheme of Urban Land Space

Urban land space has a high land development intensity, contains most of the city’s population, which is highly concentrated, especially in the urban centre of the megalopolis. According to the data from the Seventh national census of Shanghai, although the population density in the urban centre has decreased, it will still be as high as 23,100 persons/km^2^ in 2020. As a point of reference, the population density of Mumbai, Seoul, the Tokyo Metropolitan Area, and New York City in 2018, is 20,700 persons/km^2^, 15,800 persons/km^2^, 14,500 persons/km^2^ and 10,800 persons/km^2^, respectively [41]. It is important to consider the 2017 population densities of the 28,900 persons/km^2^, 32,000 persons/km^2^, and 34,000 persons/km^2^ in Jing’an, Huangpu, and Hongkou districts in the urban centre, and in combination with the requirement that the new town population density of the Shanghai urban master plan (2017–2035) reaches more than 12,000 persons/km^2^. It is also relevant to consider the different land carrying capacity of urban land space between the urban centre, the suburbs and the new towns construction of the suburbs. According to the evaluation grades of land carrying capacity in the urban centre, inner suburbs and outer suburbs (Table 6), this paper selects 20,000–25,000 and 25,000–30,000 persons/km^2^ for the suitable population carrying capacity for the “Good” and “Better” grades in the urban centre. It has also selected the value of 9000 persons/km^2^ to be the suitable population carrying capacity of the benchmark or “Common” grade in the inner suburbs. Furthermore, the carrying population per square kilometer for the other four grades is determined from high to low as 12,000–15,000, 9000–12,000, 6000–9000, and 3000–6000 persons, respectively. Simultaneously, the paper has set the values of 4500–6000 and 3000–4500 persons/km^2^ to represent the suitable population carrying capacity of the “Inferior” and “Poor” grades in the outer suburbs. Therefore, the threshold standards of the carrying population per unit area for the different grades and regions in the urban land space are obtained (Table 7).

##### Estimation Scheme of Agricultural Land Space

Agricultural land space is predominantly cultivated land, including garden and other types of agricultural land. Its primary function is providing agricultural products. Compared to urban land space, agricultural land space has less population. In this paper, the population estimation scheme is constructed by examining the population that can be carried by the food produced in the agricultural land space. According to the statistical calculations, the average food yield per unit area in the nine suburbs of Shanghai in 2017 was approximately 7800 kg/hm^2^. The highest value of 8700 kg/hm^2^ was measured in the Songjiang district, while the lowest was 7100 kg/hm^2^ in the Jiading district. According to the international food security standard line of 400 kg/person, and the present calculations, the maximum population that can be carried per square kilometer of agricultural land space is valued at 2175, while the minimum is 1775, and the average is 1950. Considering regional differences and technological development, 1900 persons/km^2^ are selected as the “Common” grade of agricultural land space that can be carried, and the other four grades are determined as 2100–2300 persons/km^2^, 1900–2100 persons/km^2^, 1700–1900 persons/km^2^, and 1500–1700 persons/km^2^, respectively.

##### Estimation Scheme of Ecological Land Space

Ecological land space maintains the regional ecological balance and sustainable development. Furthermore, it continuously provides ecological services and ensures the uninterrupted progress of social and economic activities. Previous research conducted in 2018 illustrates that China’s important ecological space area was 7.564 million km^2^ [42]. Furthermore, the per capita important ecological space was calculated to be about 0.54 hm^2^, combining with the world average production of ecological space of 1.84 hm^2^/person for the existing quality of life in China proposed by Xie et al. [43]. Taking the per capita amount as a reference, the population that can be carried per square kilometer of ecological land space is 54 persons and 185 persons. In order to facilitate calculation and grade division, the carrying population of ecological land space is set to 60–180 persons/km^2^. Furthermore, the median value of 120 persons/km^2^ is taken as the carrying population quantity of the “Common” grade, with the other four grades determined as 150–180 persons/km^2^, 120–150 persons/km^2^, 90–120 persons/km^2^ and 60–90 persons/km^2^, respectively.

#### 3.2.3. Estimation of Gross Population Carrying Capacity

According to the population carrying capacity quantitative standards of different grades and regions determined in different land spaces, the statistics area of corresponding grades (Table 5 and Table 6), and the basic idea of estimation, the population carrying capacity intervals of different grades in different land spaces are calculated by summarizing the population carrying capacity of each grade. The study concludes that the population carrying capacity in urban land space is 20,804.19–27,049.33 thousand persons. In agricultural land space, the capacity is 4450.52–4727.03 thousand persons, and in ecological land space is calculated at 221.90–270.91 thousand persons. In total, the population carrying capacity of the three types of land spaces is 25,476.61–32,047.27 thousand persons (Table 8).

In terms of the carrying population quantity of the three types of land spaces, urban land space is the most for carrying population in Shanghai. It accounts for 81.6–84.4% of the total population. It is followed by agricultural land space, which accounts for 14.7–17.5%. Lastly, the carrying population quantity in ecological land space is small, accounting for only 0.9%. Urban land space has the largest carrying population quantity, reaching 10,936.35–14,008.32 thousand persons in the inner suburbs (Table 7) and accounting for about 52% of the total population in urban land spaces. Furthermore, the urban land area in the urban centre is 275.545 km^2^, accounting for only 9.1% of the total space. However, it has a high population quantity, accounting for 26.9–28.4% of the total population of the space, with the highest population carrying density of 21,440–26,440 persons/km^2^, about 2.96–3.12 times of the space. Although the outer suburbs have a higher area of urban land space, it carries relatively lower population quantity.

### 3.3. Potential Analysis of Population Carrying Capacity

Based on the grade results of the three types of land space and the Shanghai administrative division data, the carrying capacity evaluation grade areas of the three types of land spaces in each district are extracted and calculated by using GIS software. The land population carrying capacity of each district in Shanghai is calculated and summarized with respect to the determined population carrying capacity standards per unit area of different regions and the grades in different land spaces. In addition, the potential spaces of land population carrying capacity for each district are calculated with respect to the statistical resident population of each district in Shanghai in 2017 and the evaluation of the land population carrying capacity for each district. Finally, the appropriate potential of each district is calculated when the median value of the population carrying capacity estimation interval of each district is taken to be the appropriate carrying population. The results are provided in Table 9.

In terms of the carrying population estimation, the inner suburbs have a larger population quantity. They are followed by the outer suburbs. In addition, although the urban centre has a larger population density, their quantity is the smallest due to its smaller land space area. The population carrying quantity of each district has a large variability. The Pudong New, Minhang, Chongming, and Baoshan districts exhibit a larger population carrying quantity, all larger than two million persons. Conversely, the Huangpu, Hongkou, and Changning districts in the urban centre have a smaller population carrying quantity, all smaller than one million persons.

With respect to the population carrying potential of each district, if the calculations were done according to the lower limit of the population estimation, the population of other districts in the urban centre, with the exception of Changning district, would have exceeded the population carrying capacity in 2017. More specifically, the Huangpu, Hongkou, and Putuo districts exceeded the estimation population by 39.89%, 38.16%, and 18.64%, respectively. Furthermore, the Songjiang, Qingpu, and Pudong New districts in the suburbs also exceeded the lower limit of the estimation population. If the calculations were done according to the upper limit of the population estimation, the population of the Hongkou and Huangpu districts in the urban centre and the Songjiang district in the outer suburbs would have overloaded, while other districts have a certain population carrying potential. Among the latter, the Chongming, Pudong New, and Jiading districts in the suburbs have a large population carrying potential of more than 1000 thousand persons, which are 2161.66, 2079.19, and 1105.84 thousand persons, respectively. If the calculations were done according to the appropriate carrying population, with the exception of Changning and Xuhui districts, all other districts in the urban centre would have different numbers of population overload, while other districts in the suburbs would have a certain number of population carrying potential except Songjiang district, and Chongming district would have the greatest population carrying potential, followed by the Pudong New, Fengxian, and Jiading districts. Overall, Shanghai still has a population carrying potential of 1293.30–7863.97 thousand persons, while the appropriate population carrying potential also has 4578.64 thousand persons. The population carrying potential is predominantly distributed in the suburbs, of which the outer suburbs are larger than the inner ones. On the other hand, the population in the urban centre is close to the upper limit of the estimation population carrying capacity and exceeds the appropriate population carrying capacity of 302.64 thousand persons, making the population carrying pressure larger.

## 4. Discussion

In recent decades, the estimation and prediction of the population that Shanghai can carry have never stopped. Early, the population expert Wu concluded that if the urbanized area of Shanghai can be expanded to 2113 km^2^, the total population capacity of the city can be close to 30 million [44], and in fact, the area of urban land space already reached 3028.6 km^2^ in 2017, according to his inference, Shanghai can carry more people. If the overall efficiency, including economic efficiency, social life, resource background and ecological environment can meet the development strategy and strength requirements of Shanghai, then the maximum population carrying capacity will reach 25.7 million persons in 2020 [45]; Zhang et al. adopted the probability-satisfaction method to predict the population carrying capacity of Shanghai and the result showed that the overall population carrying capacity is between 20.35 million and 30.12 million in 2020 when the probability-satisfaction level reached the acceptable level, people by the multifactor analysis [11]; Wang et al. predicted that population carrying capacity of Shanghai will reach 34.317 million people in 2050 in combination with population mortality, birth rate, and the population migration rate [46]; Based on the economic growth model, Yang et al. established a model to predict that the resident population of Shanghai may increase to 30.69 million people in 2040 [47]. The above research results are estimated or predicted under certain assumptions or scenarios, and the estimation and prediction of the population is consistent with this paper’s estimation threshold of land population carrying about 25,476.61–32,047.27 thousand people in Shanghai. In comparison, according to the internal differences of the carrying capacity of different land spaces, different estimation schemes are adopted, so that the estimation method is relatively reasonable, and the reliability of the estimation results is high in this paper.

Population regulation is an important basic project related to land carrying capacity. The Shanghai Urban Master Plan (2017–2035) requires that the resident population will be controlled at about 25 million by 2035, and the plan also proposes to explore and improve the multi-scenario planning strategy in order to regulate the matching relationship between population and land scale. From the estimation results of this paper, the population of the urban centre area exceeds the lower limit of the population carrying capacity estimation, while the suburbs still have a large population carrying potential. The results of carrying capacity evaluation and potential analysis can be used as a planning instrument of Shanghai, especially for districts with overloaded population and large population carrying potential. To this end, Shanghai should continue to further promote the layout of population and industrial development, guide the transfer and concentration of population to the suburb new towns and towns through industrial redistribution, gradually enhance the scale and intensity of population and industry carrying capacity of secondary central towns in the suburbs, constantly optimize the population spatial layout, and reduce the pressure on population carrying capacity of the urban centre area. At the same time, the new towns will play its role as the main battlefield and reservoir for Shanghai to attract talents and gather population, the government should strengthen policy and diversified housing support, improve resource allocation, perfect the housing rental system and push on the financing and supply of affordable rental housing and promote the planning standard that the population density of the new towns is not less than 12,000 persons/km^2^. In addition, Shanghai should adjust the direction of infrastructure investment, enlarge the construction of infrastructure and public service facilities in the suburbs, strive to promote the equalization of urban and suburb public services, and enhance the population carrying potential of suburb towns.

The proposed estimation method can effectively reveal and highlight the functional diversity of urban land spaces. Furthermore, it is capable of determining the spatial differences of land carrying capacity and potential of different regions through subspace and grading evaluation and population carrying capacity estimation of different regions and grades in the land spaces. However, we also note that the population carrying standards per unit area of different grades in the different land spaces have a certain impact on the total population estimation results. According to the different land spaces, combined with existing research results, planning and population density of the international megalopolis, the paper determines the different estimation schemes of population carrying standard per unit area of different grades in the land spaces, so as to reduce the impact on the estimation results. However, this study did not discuss the sensitivity of the change of carrying capacity standard to the total population estimation in Shanghai. In the future, we need to use some methods to analyze the rationality and sensitivity of the standard of carrying capacity per unit area.

## 5. Conclusions

The present paper selects Shanghai as the object of analysis and is based on the concept of the moderate scale resilient city and the idea of territorial spatial division. It employs GIS technology for land carrying capacity evaluation, population estimation, and potential analysis of the different land spaces under the constraints of natural resources and socio-economic development. The present study may also provide reference for Shanghai in improving its multi-scenario planning strategy. Furthermore, it may aid current research ideas and methods of land carrying capacity, and thus promote the research concerned with the human-land relationship.

The evaluation results indicate that the spatial difference of land carrying capacity in each land space is apparent. In urban land spaces, the evaluation index of land carrying capacity exhibits a decreasing trend from the urban centre across the inner suburbs and to the outer suburbs. Furthermore, the urban centre includes only the “Good” and “Better” grades, while the suburbs are primarily graded as “Inferior” and “Poor”. Secondly, for agricultural land space, land carrying capacity of the outer suburbs is generally taken to be higher than that of the inner suburbs. Lastly, for ecological land space, the state index of the carrying capacity evaluation shows a descending trend from the outside to the inside.

The estimation results indicate that the total population of land carrying capacity in Shanghai is 25,476.61–32,047.27 thousand people. More specifically, urban land space can carry 20,804.19–27,049.33 thousand people, which accounts for 81.6–84.4% of the total carrying population, indicating it is the main land space for population carrying capacity in Shanghai. The inner suburbs carry the highest population quantity, especially the urban land space in the inner suburbs. Furthermore, although the outer suburbs have a larger area of urban land space, it has a relatively low population quantity. On the other hand, the urban centre area is small, but it has a large population carrying density.

Based on the resident population number of Shanghai in 2017, the city still has an overall population carrying potential of 1293.30–7863.97 thousand persons and the appropriate population carrying potential of 4578.64 thousand persons. Furthermore, there are great spatial differences in the land population carrying potential. The population in the urban centre area exceeds the lower limit of the population carrying capacity estimation and the appropriate population carrying capacity, especially in the Hongkou and Huangpu District. In addition, the suburbs, especially the outer ones, have a larger population carrying potential.

Overall, although population growth may affect resources availability and ecological environment quality, resource scarcity and environmental capacity will also restrict population growth, especially land and water resources, and the environmental quality in large cities. Simultaneously, socio-economic and technological development and government behavior will enhance land use intensity, increase economic output, strengthen urban infrastructure construction and improve water, soil and air quality. In turn, these changes will affect the urban land population carrying capacity. Because of this, urban land carrying capacity is considered to be dynamic, while the method proposed in the paper may be used for continuous dynamic updating, thus providing a typical case reference for similar studies of other large cities and urban agglomerations.

## Figures and Tables

**Figure 1 ijerph-19-08240-f001:**
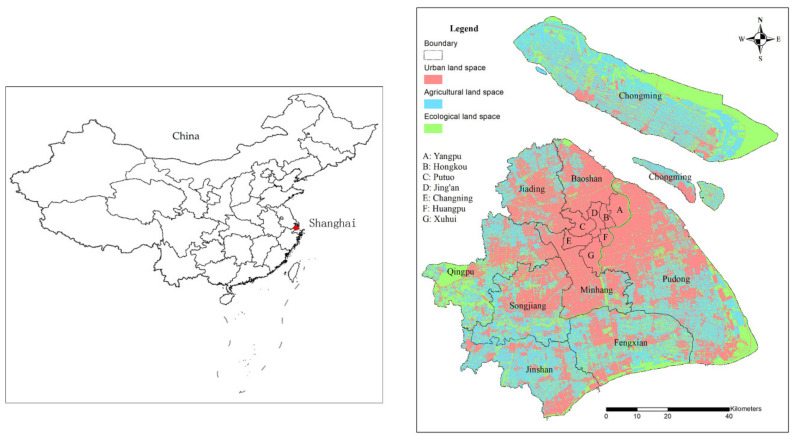
Distribution of the three types of land spaces in Shanghai.

**Figure 2 ijerph-19-08240-f002:**
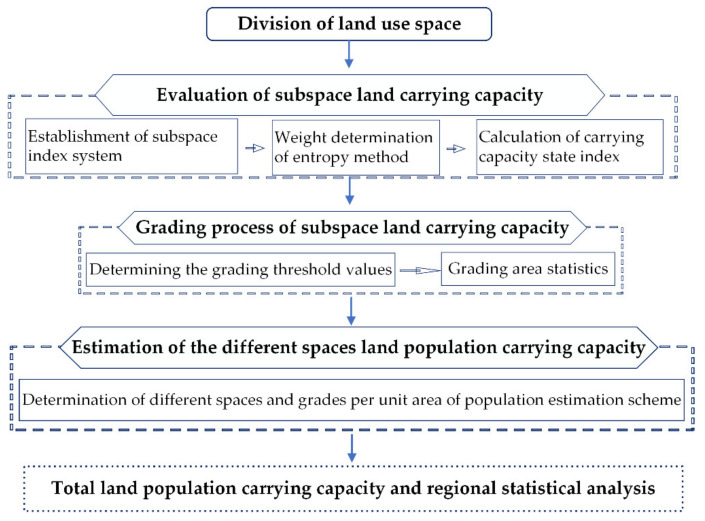
Estimation process of land population carrying capacity based on GIS.

**Figure 3 ijerph-19-08240-f003:**
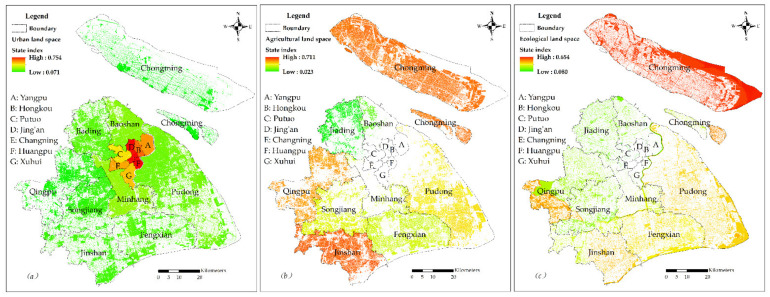
The carrying capacity evaluation values of the three subspaces in Shanghai. (**a**) the carrying capacity state index of urban land space; (**b**) the carrying capacity state index of agricultural land space; (**c**) the carrying capacity state index of ecological land space.

**Figure 4 ijerph-19-08240-f004:**
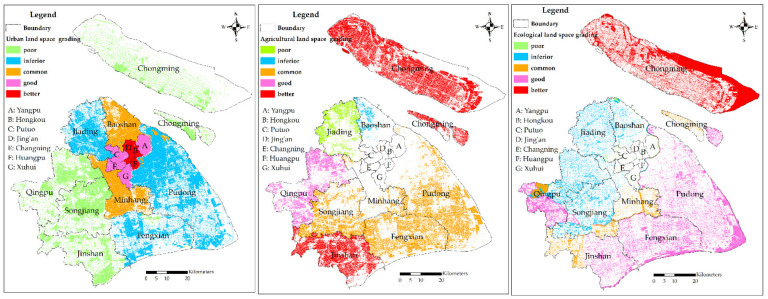
Carrying capacity grades of the three subspaces in Shanghai.

**Table 1 ijerph-19-08240-t001:** Data source.

Data Types	The Data Source
Statistics data	Shanghai Statistical Yearbook in 2018
	Statistical Bulletin of Districts in 2018
	Shanghai water resources bulletin in 2017
	Environmental quality status bulletin of districts in 2017
Spatial data	Landsat8_OLI (Geospatial Data Cloud, http://www.gscloud.cn/, accessed on 5 January 2021)
	DEM data (Geospatial Data Cloud, http://www.gscloud.cn/, accessed on 5 January 2021)
	Shanghai administrative divisions map
	Shanghai Traffic Map in 2017
	Shanghai Land Use Status Database in 2017
	Shanghai comprehensive evaluation map of surface soil environmental quality
	Comprehensive evaluation map of water quality districts of Shanghai in 2017 Contour Map of Shanghai Land Subsidence
	Shanghai suitability zoning map of natural foundation engineering construction
	Shanghai potential geological disaster zoning map

**Table 2 ijerph-19-08240-t002:** The current land use classification and codes in China (GB/T21010-2017).

Three Categories	Names and Codes of Level I Types	Names and Codes of Level II Types
Agricultural land	Arable land (01)	Paddy field (011), watered land (012), dry land (013)
	Garden land (02)	Orchard (021), tea garden (022), other garden land (023)
	Woodland (03)	Forested land (031), shrub land (032), other woodland (033)
	Grassland (04)	Natural pasture (041), artificial pasture (042)
	Transportation land (10)	Rural road land (104)
	Water area and water conservancy facilities land (11)	Pond (114), ditch (117)
	Other land (12)	Agricultural facilities land (122), ridge of field (123)
Construction land	Commercial land (05)	Wholesale and retail land (051), accommodation and catering land (052), commercial and financial land (053), other commercial land (054)
	Mining warehouse land (06)	Industrial land (061), mining land (062), warehouse land (063)
	Residential land (07)	Urban residential land (071), rural housing land (072)
	Public management and service land (08)	Government organization land (081), press and publication land (082), science and education land (083), medical and health charity land (084), sport and entertainment land (085), public facilities land (086), park and green space (087), scenic spot and facilities land (088)
	Special land (09)	Military facilities land (091), embassies and consulates land (092), educational supervision site land (093), religious land (094), funeral land (095)
	Transportation land (10)	Railway land (101), highway land (102), streets and alleys land (103), airport land (105), port and wharf land (106), pipeline transportation land (107)
	Water area and water conservancy facilities land (11)	Reservoir (113), hydraulic construction land (118)
	Other land (12)	Vacant land (121)
Unused land	Water area and water conservancy facilities land (11)	River (111), lake (112), tidal flat (115), inland tidal flat (116), glaciers and permanent snow (119)
	Grassland (04)	Other grassland (043)
	Other land (12)	Saline land (124), marshland (125), sand land (126), bare land (127)

**Table 3 ijerph-19-08240-t003:** The division scheme of land use space and area statistics.

Land Space Type	Major Land Use Types	Area (km^2^)
Urban land space	Commercial land, mining warehouse land, residential land, transportation land, public management and service land, special land, other land (vacant land), water area and water conservancy facilities land (hydraulic construction land)	3028.631
Agricultural land space	Arable land, garden land, rural road land, other land (agricultural facilities land, ridge of field)	2265.151
Ecological land space	Woodland, grassland, water area and water conservancy facilities land and other land (sand, bare land)	1777.911

Note: The total area does not include the Yangtze River and Hangzhou Bay water area.

**Table 4 ijerph-19-08240-t004:** Evaluation indicator systems for three subspaces and their weights.

Urban land space	Indicators	Per capita Construction Land	Industrial Land Proportion	Population Density	Economic Density	Urbanization Rate
	Weights	0.055	0.066	0.154	0.20	0.030
	Indicators	Hospital beds per 10,000 people	Road network density	Infrastructure investment per unit area	Suitability of natural foundation engineering construction	Land subsidence
	Weights	0.142	0.104	0.183	0.033	0.033
Agricultural land space	Indicators	Per capita cultivated land	per capita food occupancy	Soil Environmental Quality	Output value per unit agricultural land	Agricultural labor productivity
	Weights	0.168	0.182	0.030	0.164	0.210
	Indicators	Cultivated land concentration	Slope			
	Weights	0.175	0.071			
Ecological land space	Indicators	Geological hazards susceptibility	Comprehensive water quality Index	Per capita park green area	Air quality index	PM2.5
	Weights	0.057	0.320	0.105	0.039	0.052
	Indicators	Vegetation coverage	wetland area ratio			
	Weights	0.238	0.189			

**Table 5 ijerph-19-08240-t005:** The interval values and areas of grades in the three subspaces.

Grade	Urban Land Space		Agricultural Land Space		Ecological Land Space	
	Threshold Value	Area (km^2^)	Threshold Value	Area (km^2^)	Threshold Value	Area (km^2^)
Poor	<0.154	950.943	<0.444	125.791	<0.192	18.472
Inferior	[0.154, 0.218)	1268.640	[0.444, 0.484)	32.773	[0.192, 0.267)	286.183
Common	[0.218, 0.272)	532.118	[0.484, 0.597)	882.594	[0.267, 0.363)	144.087
Good	[0.272, 0.572)	197.620	[0.597, 0.646)	205.977	[0.363, 0.510)	721.106
Better	≥0.572	79.310	≥0.646	1018.017	≥0.510	608.063
Total (km^2^)	/	3028.631	/	2265.151	/	1777.911

**Table 6 ijerph-19-08240-t006:** Area statistics of different regional grades of carrying capacity in urban land space (km^2^).

Grade	Urban Land Space			
	Urban Centres	Inner Suburbs	Outer Suburbs	Total
Poor	0	0.280	950.662	950.943
Inferior	0	1022.329	246.311	1268.640
Common	0	532.118	0	532.118
Good	196.235	1.385	0	197.6203
Better	79.310	0	0	79.310
Total	275.545	1556.113	1196.973	3028.631

**Table 7 ijerph-19-08240-t007:** Estimation standards per unit area and population carrying capacity in urban land space.

Grade	Urban Centres		Inner Suburbs		Outer Suburbs	
	Carrying Density (Persons/km^2^)	Carrying Population (1000 Persons)	Carrying Density (Persons/km^2^)	Carrying Population (1000 Persons)	Carrying Density (Persons/km^2^)	Carrying Population (1000 Persons)
Poor	0	0	3000–6000	0.84–1.68	3000–4500	2851.99–4277.98
Inferior	0	0	6000–9000	6133.98–9200.96	4500–6000	1108.40–1477.86
Common	0	0	9000	4789.06	0	0
Good	20,000–25,000	3924.70–4905.87	9000–12,000	12.47–16.62	0	0
Better	25,000–30,000	1982.75–2379.30	12,000–15,000	0	0	0
Total		5907.45–7285.17		10,936.35–14,008.32		3960.39–5755.84

**Table 8 ijerph-19-08240-t008:** Results of population carrying capacity estimation (thousand persons).

Grade	Space Types			
	Urban Land Space	Agricultural Land Space	Ecological Land Space	Total
Poor	2852.83–4279.66	188.69–213.84	1.11–1.66	3042.63–4495.16
Inferior	7242.38–10,678.82	55.71–62.27	25.76–34.34	7323.85–10,775.43
Common	4789.06	1676.93	17.29	6483.28
Good	3937.17–4922.49	391.36–432.55	86.53–108.17	4415.06–5463.21
Better	1982.75–2379.30	2137.83–2341.44	91.21–109.45	4211.79–4830.19
Total	20,804.19–27,049.33	4450.52–4727.03	221.90–270.91	25,476.61–32,047.27

**Table 9 ijerph-19-08240-t009:** Population carrying capacity and potential of each district in Shanghai (thousand persons).

District Name		Estimation Population Carrying Capacity	Resident Population in 2017	Potential Population Carrying	Appropriate Carrying Population	Potential Appropriate Population Carrying
Urban centres	Xuhui	1028.71–1285.90	1088.3	−59.59–197.60	1157.31	69.01
	Yangpu	1115.95–1394.11	1313.4	−197.45–80.71	1255.03	−58.37
	Putuo	1076.96–1346.23	1284.7	−207.74–61.53	1211.60	−73.11
	Hongkou	578.29–693.95	799.	−220.71–−105.05	636.12	−162.88
	Huangpu	468.06–561.69	654.8	−186.74–−93.11	514.88	−139.93
	Jing’an	915.75–1098.90	1066.2	−150.45–32.70	1007.33	−58.87
	Changning	724.63–905.80	693.7	30.93–212.10	815.21	121.51
Subtotal		5908.35–7286.59	6900.1	−991.76–386.49	6597.46	−302.64
Inner suburbs	Pudong	5355.55–7607.59	5528.4	−172.85–2079.19	6481.57	953.17
	Jiading	1838.25–2687.64	1581.8	256.45–1105.84	2262.95	681.15
	Minghang	2636.93–2641.39	2534.3	102.63–107.09	2639.16	104.86
	Baoshan	2166.19–2176.27	2030.8	135.39–145.47	2171.23	140.43
Subtotal		11,996.92–15,112.89	11,675.3	321.62–3437.59	13,554.90	1879.60
Outer suburbs	Fengxian	1666.62–2049.67	1155.3	511.32–894.37	1858.15	702.85
	Qingpu	1092.17–1477.74	1205.3	−113.13–272.44	1284.96	79.66
	Songjiang	1243.74–1671.40	1751.3	−507.56–−79.90	1457.57	−293.73
	Jinshan	1232.68–1592.72	801.4	431.28–791.32	1412.70	611.30
	Chongming	2336.13–2856.26	694.6	1641.53–2161.66	2596.20	1901.60
Subtotal		7571.34–9647.79	5607.9	1963.44–4039.89	8609.57	3001.67
Total		25,476.61–32,047.27	24,183.3	1293.30–7863.97	28,761.94	4578.64

## Data Availability

The data presented in this study are available on request from the corresponding author.

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
