# Peer review of "Estimation and Potential Analysis of Land Population Carrying Capacity in Shanghai Metropolis"

_ijerph, 2022, doi:10.3390/ijerph19148240_

Round 1

Reviewer 1 Report

I suggest framing the  topic in an international policy context beyond the reference to the Shanghai Urban Master Plan (2017-2035), for example, by linking it to the United Nations Sustainable Development Goal 11. This would lend support and greater robustness to the indicators formulated in Table 3.

Regarding section 2.1, I would first consider the list of data used without further elaboration (table 1 ) and then explain how additional spatial information was obtained (otherwise the linkage between the formulas and the table is not clear). Moreover, tables or figures should be inserted immediately after the citation in the text.

The contrast of figure 4 needs to be improved because it is not readable.

Regarding the section “3.2. Estimation of land population carrying capacity” , it is not clear whether it is methods or results.

In Table 8; column 4. Replace Potential population carrying      or      Population carrying (potential)

The same for “potential” in column 6.

Author Response

Dear Reviewer

Thank you very much for your letter and the referees’ reports. All of your suggestions and comments for the manuscript are very important, they have an important guiding significance for my paper writing and research work. Here below is our description on revision according to your comments and suggestions, and the modified contents are marked by using the “Track Changes” function in the revised manuscript.

Q1. I suggest framing the topic in an international policy context beyond the reference to the Shanghai Urban Master Plan (2017-2035), for example, by linking it to the United Nations Sustainable Development Goal 11. This would lend support and greater robustness to the indicators formulated in Table 3.

Answer:

Your suggestions are very good. We have added some expressions of an international policy context in the third paragraph of the introduction in the revised manuscript. In addition, the topic of the paper is determined based on the concept of moderate scale resilient city and combined with the Shanghai Urban Master Plan (2017–2035). The plan is consistent with the concept of the United Nations sustainable development goal 11. The targets are also to optimize urban space, control urban scale, strengthen ecological environment protection,ensure the safe operation of the city and improve the management system under the constraints of resources and environment, constantly improve the adaptability and resilience of the city and make cities more sustainable. Therefore, the selection of indicators is also based on the concept, and the indicator system is based on the characteristics of the resource background elements, social and economic development and the problems faced by Shanghai. It should be said that the indicator system is relatively scientific. Of course, some indicators such as protecting the world's cultural and natural heritage have not been considered, and we will also consider them in future research. The revision context is as follow:

At the same time, we also note that the sustainable development goal 11 of the United Nations mainly focuses on cities and puts forward the goal of the safety, resilience and sustainability of cities and human settlements to tackle the challenges of environmental and other disasters. However, with rapid urbanization and urban agglomeration development, the scale of cities has expanded drastically,……

Q2. Regarding section 2.1, I would first consider the list of data used without further elaboration (table 1) and then explain how additional spatial information was obtained (otherwise the linkage between the formulas and the table is not clear). Moreover, tables or figures should be inserted immediately after the citation in the text.

Answer:

First of all, thank you for your comments. Some index data is processed based on the spatial data, so we think that some explanations should be added appropriately. Secondly, we also modified the table order, paragraph structure and local statements according to your suggestions in the revision.

Q3. The contrast of figure 4 needs to be improved because it is not readable.

Answer:

Thanks, according to the suggestions,all figures have been reprocessed to meet the resolutions and readability requirement in the revised manuscript.

Q4. Regarding the section “3.2. Estimation of land population carrying capacity”, it is not clear whether it is methods or results.

Answer:

Your comment is correct. The section 3.2 is the evaluation results, so we have changed the title of 3.2 to “Estimation results of land population carrying capacity ''.

Q5. In Table 8; column 4. Replace Potential population carrying      or      Population carrying (potential) ;The same for “potential” in column 6.

Answer:

Thanks for your suggestions, we have replaced column 4 and column 6 in Table 8 with potential population carrying and potential appropriate population carrying, respectively.

Thank you for your suggestions and comments again.

Sincerely yours,

Hefeng Wang

Reviewer 2 Report

Page 3 of 17:

 ……by slope calculation in the ArcGIS software.

Do not give the names of such programs. It would be more accurate to specify it as any GIS software. If ArcGIS offers you a free service, you can use it in the "acknowledge" section at the end of the article, not in the article.

Page 4 of 17

Formula 2

If you have created/found all the formulas in the article for the first time in the literature or if you are using them in such a study, please write this situation in the article. If not, please cite them from previous work.

Page 5 of 17

 ……..12 Level I types and 56 Level II subtypes [34,35].

Please see Figure 8 in the study "Zoning plan-based legal confiscation without expropriation in Turkey in light of ECHR decisions" in Web of Science. Visually grade spatial plans in this way in China and give this new figure in the article. So that all readers can easily understand the Chinese spatial planning in a way.

Table 2. the division scheme of land use space and area statistics

Examine figure 4 in the article "Determining the property ownership on cadastral works in Turkey" in Web of Science. Prepare a figure similar to this one about land acquisition/ownership in China and insert this section. Thus, readers can more easily understand the relationship between population carrying capacity and immovable property.

While doing this, it will be sufficient to consider the major land use types you have given in Table 2. According to the information you will give here, in the results section, if there is an overpopulation in a city or there will be an overpopulation in that region in the future, based on your article estimation method is necessary to analyse the measures to be taken against this, with the property situation in that region.

Page 8 of 17

Figure 2.

the resolution of the figure is not enough; it will be better to rearrange it.

Page 9 of 17

Figure 3.

Page 10 of 17

Figure 4.

The resolutions/readability of the figures and legend information are not sufficient; they should be uploaded to the journal system with high output quality on the GIS platform. And they should be changed within the article as well.

4. Discussion and Conclusions

The word "discussion" should be removed from this thread. Discussion is a topic that can be used if there is a comparison of the findings in the articles that have done similar studies before with your findings. For this reason, it will suffice to give the title of results only for this section containing your results.

1st para.

According to what is stated in this paragraph, human settlement in any area should be based on the previous spatial planning of that area. however, before this planning, it is also necessary to estimate the number of people expected to live in the area. Is the population projection taken into account when planning and using land in accordance with that plan in the areas that you have considered as the study area in this article? And is this situation audited? Please provide information about this in the results section. If the number of people living in the study area in accordance with the population projection envisaged in the plan, why is there a need for such a study/your work? If not, is there a problem with the permit process regarding the accommodation or work in the work area of ​​overcrowded people who do not comply with the plan? What kind of measures is the Chinese government taking in this regard? Please evaluate and answer the above information and indicate this situation in the relevant paragraph.

2nd para. and 3th paragraph.

In this section, I expect you to evaluate the widespread impact of the study for China in general and to consider its applicability to potential readers' home countries. that is, how useful would it be to make an estimation of the possible population density in the planned areas with this study? Please look into this situation. and address this situation both with the dissemination of the study throughout the country of china. At the same time, highlight the issues that the scientists who will read your article should consider when they want to apply similar work in different countries.

Author Response

Dear Reviewer

Thank you very much for your letter and the referees’ reports. All of your suggestions and comments for the manuscript are very important, they have an important guiding significance for my paper writing and research work. Here below is our description on revision according to your comments and suggestions, and the modified contents are marked by using the “Track Changes” function in the revised manuscript.

Q1. Page 3 of 17:

 ……by slope calculation in the ArcGIS software.

Do not give the names of such programs. It would be more accurate to specify it as any GIS software. If ArcGIS offers you a free service, you can use it in the "acknowledge" section at the end of the article, not in the article.

Answer:

Thank you for your suggestions. We have modified the expression throughout the revised manuscript.

Q2. Page 4 of 17

Formula 2

If you have created/found all the formulas in the article for the first time in the literature or if you are using them in such a study, please write this situation in the article. If not, please cite them from previous work.

Answer:

Thank you for your question. Your comment is very correct, and we have added references in the manuscript as follows:

The normalized differential vegetation index (NDVI) of Shanghai was calculated by using equation (2) with remote sensing software [34]. Based on the this, the vegetation coverage of Shanghai was calculated by Equation (3) [35].

……

In the paper, the = 0.7 and  = 0.05 for calculation. If > 0.7, it is assumed that full vegetation coverage = 1, and if < 0.05, it is assumed that bare ground = 0[36].

  1. Laura, U.; Caroline, N. ; Karl, H. ; David, L. ; Elizabeth, M.; Alexei, L.; Lvan, M.; Janne, L.; Albert, P.C. Detecting inter-annual variations in the phenology of evergreen conifers using long-term modis vegetation index time series. Remote Sens-ing, 2017, 9(1):49. doi:10.3390/rs9010049
  2. Carlson, T.N.; Ripley,D.A. On the relation between NDVI, fractional vegetation cover and leaf area index. Remote Sensing of Environment, 1997, 62:241-252. doi: 10.1016/S0034-4257(97)00104-1
  3. Qin, Z. H.; Li, W. J.; Xu, B.; Zhang, W. C. Estimation method of land surface emissivity for retrieving land surface temperature from Landsat TM6 data. Advances in Marine Science, 2004,22(suppl.),129-137.(in Chinese).

Q3. Page 5 of 17

 ……..12 Level I types and 56 Level II subtypes [34,35].

Please see Figure 8 in the study "Zoning plan-based legal confiscation without expropriation in Turkey in light of ECHR decisions" in Web of Science. Visually grade spatial plans in this way in China and give this new figure in the article. So that all readers can easily understand the Chinese spatial planning in a way.

Table 2. the division scheme of land use space and area statistics

Examine figure 4 in the article "Determining the property ownership on cadastral works in Turkey" in Web of Science. Prepare a figure similar to this one about land acquisition/ownership in China and insert this section. Thus, readers can more easily understand the relationship between population carrying capacity and immovable property.

While doing this, it will be sufficient to consider the major land use types you have given in Table 2. According to the information you will give here, in the results section, if there is an overpopulation in a city or there will be an overpopulation in that region in the future, based on your article estimation method is necessary to analyse the measures to be taken against this, with the property situation in that region.

Answer:

Thank you for your comments and the references provided. Your suggestions may be right, but we consider division of land use space based on the "Technical specifications and preparation guidelines of the master plan for eco-nomic and social development of cities and counties (for Trial Implementation)" and the "Current Land Use Classification Standard", and it is more divided from the perspective of main function according to land use codes of different level types. In order to make readers understand the division idea of land use spaces, we have added Table 2 of the current land use classification and codes in the revised version to help readers understand the merging scheme of land use types in different land spaces in Table 3.

This study discussed the estimation of land population carrying capacity and potential analysis and did not consider the relationship between specific population carrying capacity and real estate. Your suggestions provide us with ideas for further research. Of course, in the second paragraph of discussion section, we also put forward relevant measures to alleviate the population pressure in the urban centre, and put forward simple real estate measures for the new town construction.

In addition, it should be noted that from the perspective of planning levels, China's territorial spatial planning is divided into five levels, corresponding to China's administrative system, namely, the national level, provincial level, municipal level, county level and township level; From the perspective of planning contents, territorial spatial planning can be divided into three types: overall planning, detailed planning and relevant special planning.

Q4. Page 8 of 17

Figure 2.

the resolution of the figure is not enough; it will be better to rearrange it.

Page 9 of 17

Figure 3.

Page 10 of 17

Figure 4.

The resolutions/readability of the figures and legend information are not sufficient; they should be uploaded to the journal system with high output quality on the GIS platform. And they should be changed within the article as well.

Answer:

Figure2, Figure 3 and Figure 4 have been reprocessed to meet the resolutions and readability requirement in the revised manuscript.

Q5.  4. Discussion and Conclusions

The word "discussion" should be removed from this thread. Discussion is a topic that can be used if there is a comparison of the findings in the articles that have done similar studies before with your findings. For this reason, it will suffice to give the title of results only for this section containing your results.

1st para.

According to what is stated in this paragraph, human settlement in any area should be based on the previous spatial planning of that area. however, before this planning, it is also necessary to estimate the number of people expected to live in the area. Is the population projection taken into account when planning and using land in accordance with that plan in the areas that you have considered as the study area in this article? And is this situation audited? Please provide information about this in the results section. If the number of people living in the study area in accordance with the population projection envisaged in the plan, why is there a need for such a study/your work? If not, is there a problem with the permit process regarding the accommodation or work in the work area of ​​overcrowded people who do not comply with the plan? What kind of measures is the Chinese government taking in this regard? Please evaluate and answer the above information and indicate this situation in the relevant paragraph.

2nd para. and 3th paragraph.

In this section, I expect you to evaluate the widespread impact of the study for China in general and to consider its applicability to potential readers' home countries. that is, how useful would it be to make an estimation of the possible population density in the planned areas with this study? Please look into this situation. and address this situation both with the dissemination of the study throughout the country of china. At the same time, highlight the issues that the scientists who will read your article should consider when they want to apply similar work in different countries.

Answer:

Thank you for your suggestions. Firstly, we have reprocessed the section of discussion and conclusions and split into two sections. Secondly, we discussed the rationality of the estimation results in this paper by comparing and analyzing the population forecast of scholars and population control requirement of The Shanghai Urban Master Plan (2017-2035). Then we also discussed some relevant policies and measures to optimize the spatial distribution of population in Shanghai. The specific revision contents are presented in the discussion and conclusion section, especially the first and second paragraphs of the discussion section. There may be some inadequacies in the revision, we need to carry out in the future. The revision context of the first and second paragraphs in the discussion section is as follow:

In recent decades, the estimation and prediction of the population that Shanghai can carry have never stopped. Early, the population expert Wu concluded that if the urbanized area of Shanghai can be expanded to 2113km2, the total population capacity of the city can be close to 30 million [44], and in fact, the area of urban land space al-ready reached 3028.6 km2 in 2017, according to his inference, Shanghai can carry more people. If the overall efficiency, including economic efficiency, social life, resource background and ecological environment can meet the development strategy and strength requirements of Shanghai, then the maximum population carrying capacity will reach 25.7 million persons in 2020 [45]; Zhang et al. adopted the probability-satisfaction method to predict the urban population carrying capacity of Shanghai and the prediction result shown that the overall population carrying capacity is be-tween 20.35 million and 30.12 million in 2020 when the probability-satisfaction level reached the acceptable level, people by the multifactor analysis [11]; Wang et al. predicted that population carrying capacity of Shanghai will reach 34.317 million people in 2050 in combination with population mortality, birth rate, and the population migration rate [46]; Based on the economic growth model, Yang et al. established a model to predict that the resident population of Shanghai may increase to 30.69 million people in 2040 [47]. The above research results are estimated or predicted under certain assumptions or scenarios, and the estimation and prediction of the population is consistent with this paper’s estimation threshold of land population carrying about 25,476.61–32,047.27 thousand people in Shanghai. In comparison, according to the internal differences of the carrying capacity of different land spaces, different estimation schemes are adopted, so that the estimation method is relatively reasonable, and the reliability of the estimation results is high in this paper.

Population regulation is an important basic project related to land carrying capacity. The Shanghai Urban Master Plan (2017-2035) requires that the resident population will be controlled at about 25 million by 2035, and the plan also proposes to explore and improve the multi-scenario planning strategy in order to regulate the matching relationship between population and land scale. From the estimation results of this paper, the population of the urban centre area exceeds the lower limit of the population carrying capacity estimation, while the suburbs still have a large population carrying potential. The results of carrying capacity evaluation and potential analysis can be used as a planning instrument of Shanghai, especially for districts with overloaded population and large population carrying potential. To this end, Shanghai should continue to further promote the layout of population and industrial development, guide the transfer and concentration of population to the suburb new towns and towns through industrial redistribution, gradually enhance the scale and intensity of population and industry carrying capacity of secondary central towns in the suburbs, constantly optimize the population spatial layout, and reduce the pressure on population carrying capacity of the urban centre area. At the same time, the new towns will play its role as the main battlefield and reservoir for Shanghai to attract talents and gather population, the government should strengthen policy and diversified housing support, improve resource allocation, perfect the housing rental system and push on the financing and supply of affordable rental housing and promote the planning standard that the population density of the new towns is not less than 12,000 persons/km2. In addition, Shanghai should adjust the direction of infrastructure investment, enlarge the construction of infrastructure and public service facilities in the suburbs, strive to promote the equalization of urban and suburb public services, and enhance the population carrying potential of suburb towns.

Thank you for your suggestions and comments again.

Sincerely yours,

Hefeng Wang

Reviewer 3 Report

The paper is interesting and the topic may be worthy of research. The research has scientific soundness, so I think the manuscript may be considered for publication if some minor shortcomings are improved. Below I detail the issues that should be addressed by the authors:

- Fig 1: the first map must be contextualized in a wider way, so that people that are not familiar with Shanghai can fully understand it.

- The discussion is too poor and scarce. It is focused in a excessive local way which reduces its interest the international readers of IJERPH (e.g. lots of local names of districts from Shanghai not placed in maps make the text here quite confusing). In addition, bibliographic references to other studies are missing here. This section should be rewritten to enhance its academic and scientific scholarship. A more international approach should adopted citing other case studies and highlighting in which this study or methodological proposal corroborates/contradicts or improves the results of other studies (citing them). A more self criticism approach is also missing. Authors should emphasize the limitations of their proposal and which issues may be developed or improved in future lines of research. Policy implications of the study must also be highlighted.

- A specific conclusion section with short summary of the main findings obtained must be generated. The study is quite long and dense, so discussion and conclusion must be split into two sections so that readers of the journals can assess their interest in the results obtained in the study at a glance.

Author Response

Dear Reviewer

Thank you very much for your letter and the referees’ reports. All of your suggestions and comments for the manuscript are very important, they have an important guiding significance for my paper writing and research work. Here below is our description on revision according to your comments and suggestions, and the modified contents are marked by using the “Track Changes” function in the revised manuscript.

Comments and Suggestions for Authors

The paper is interesting and the topic may be worthy of research. The research has scientific soundness, so I think the manuscript may be considered for publication if some minor shortcomings are improved. Below I detail the issues that should be addressed by the authors:

Q1. - Fig 1: the first map must be contextualized in a wider way, so that people that are not familiar with Shanghai can fully understand it.

Answer:

Thank you for your suggestions. We have modified Fig 1, and the spatial position of Shanghai is shown in China.

Q2. - The discussion is too poor and scarce. It is focused in a excessive local way which reduces its interest the international readers of IJERPH (e.g. lots of local names of districts from Shanghai not placed in maps make the text here quite confusing). In addition, bibliographic references to other studies are missing here. This section should be rewritten to enhance its academic and scientific scholarship. A more international approach should adopted citing other case studies and highlighting in which this study or methodological proposal corroborates/contradicts or improves the results of other studies (citing them). A more self criticism approach is also missing. Authors should emphasize the limitations of their proposal and which issues may be developed or improved in future lines of research. Policy implications of the study must also be highlighted.

- A specific conclusion section with short summary of the main findings obtained must be generated. The study is quite long and dense, so discussion and conclusion must be split into two sections so that readers of the journals can assess their interest in the results obtained in the study at a glance.

Answer:

Thank you for your suggestions and comments, according to your suggestions, we have rewritten and split discussion and conclusion into two sections, and added other study references analysis, discussion on insufficient research method and policy implications in the discussion section. In addition, we have also simplified the conclusions in the revision. The revision context is as follow:

  1. Discussion

In recent decades, the estimation and prediction of the population that Shanghai can carry have never stopped. Early, the population expert Wu concluded that if the urbanized area of Shanghai can be expanded to 2113km2, the total population capacity of the city can be close to 30 million [44], and in fact, the area of urban land space al-ready reached 3028.6 km2 in 2017, according to his inference, Shanghai can carry more people. If the overall efficiency, including economic efficiency, social life, resource background and ecological environment can meet the development strategy and strength requirements of Shanghai, then the maximum population carrying capacity will reach 25.7 million persons in 2020 [45]; Zhang et al. adopted the probability-satisfaction method to predict the population carrying capacity of Shanghai and the result showed that the overall population carrying capacity is between 20.35 million and 30.12 million in 2020 when the probability-satisfaction level reached the acceptable level, people by the multifactor analysis [11]; Wang et al. predicted that population carrying capacity of Shanghai will reach 34.317 million people in 2050 in combination with population mortality, birth rate, and the population migration rate [46]; Based on the economic growth model, Yang et al. established a model to predict that the resident population of Shanghai may increase to 30.69 million people in 2040 [47]. The above research results are estimated or predicted under certain assumptions or scenarios, and the estimation and prediction of the population is consistent with this paper’s estimation threshold of land population carrying about 25,476.61–32,047.27 thousand people in Shanghai. In comparison, according to the internal differences of the carrying capacity of different land spaces, different estimation schemes are adopted, so that the estimation method is relatively reasonable, and the reliability of the estimation results is high in this paper.

Population regulation is an important basic project related to land carrying capacity. The Shanghai Urban Master Plan (2017-2035) requires that the resident population will be controlled at about 25 million by 2035, and the plan also proposes to explore and improve the multi-scenario planning strategy in order to regulate the matching relationship between population and land scale. From the estimation results of this paper, the population of the urban centre area exceeds the lower limit of the population carrying capacity estimation, while the suburbs still have a large population carrying potential. The results of carrying capacity evaluation and potential analysis can be used as a planning instrument of Shanghai, especially for districts with overloaded population and large population carrying potential. To this end, Shanghai should continue to further promote the layout of population and industrial development, guide the transfer and concentration of population to the suburb new towns and towns through industrial redistribution, gradually enhance the scale and intensity of population and industry carrying capacity of secondary central towns in the suburbs, constantly optimize the population spatial layout, and reduce the pressure on population carrying capacity of the urban centre area. At the same time, the new towns will play its role as the main battlefield and reservoir for Shanghai to attract talents and gather population, the government should strengthen policy and diversified housing support, improve resource allocation, perfect the housing rental system and push on the financing and supply of affordable rental housing and promote the planning standard that the population density of the new towns is not less than 12,000 persons/km2. In addition, Shanghai should adjust the direction of infrastructure investment, enlarge the construction of infrastructure and public service facilities in the suburbs, strive to promote the equalization of urban and suburb public services, and enhance the population carrying potential of suburb towns.

The proposed estimation method can effectively reveal and highlight the functional diversity of urban land spaces. Furthermore, it is capable of determining the spatial differences of land carrying capacity and potential of different regions through subspace and grading evaluation and population carrying capacity estimation of different regions and grades in the land spaces. However, we also note that the population carrying standards per unit area of different grades in the different land spaces have a certain impact on the total population estimation results. According to the different land spaces, combined with existing research results, planning and population density of the international megalopolis, the paper determines the different estimation schemes of population carrying standard per unit area of different grades in the land spaces, so as to reduce the impact on the estimation results. However, this study did not discuss the sensitivity of the change of carrying capacity standard to the total population estimation in Shanghai. In the future, we need to use some methods to analyze the rationality and sensitivity of the standard of carrying capacity per unit area.

  1. Conclusions

The present paper selects Shanghai as the object of analysis and is based on the concept of the moderate scale resilient city and the idea of territorial spatial division. It employs GIS technology for land carrying capacity evaluation, population estimation, and potential analysis of the different land spaces under the constraints of natural re-sources and socio-economic development. The present study may also provide refer-ence for Shanghai in improving its multi-scenario planning strategy. Furthermore, it may aid current research ideas and methods of land carrying capacity, and thus pro-mote the research concerned with the human-land relationship.

The evaluation results indicate that the spatial difference of land carrying capacity in each land space is apparent. In urban land spaces, the evaluation index of land carrying capacity exhibits a decreasing trend from the urban centre across the inner suburbs and to the outer suburbs. Furthermore, the urban centre includes only the “Good” and “Better” grades, while the suburbs are primarily graded as “Inferior” and “Poor”. Secondly, for agricultural land space, land carrying capacity of the outer suburbs is generally taken to be higher than that of the inner suburbs. Lastly, for ecological land space, the state index of the carrying capacity evaluation shows a descending trend from the outside to the inside.

The estimation results indicate that the total population of land carrying capacity in Shanghai is 25,476.61–32,047.27 thousand people. More specifically, urban land space can carry 20,804.19–27,049.33 thousand people, which accounts for 81.6–84.4% of the total carrying population, indicating it is the main land space for population carrying capacity in Shanghai. The inner suburbs carry the highest population quantity, especially the urban land space in the inner suburbs. Furthermore, although the outer suburbs have a larger area of urban land space, it has a relatively low population quantity. On the other hand, the urban centre area is small, but it has a large population carrying density.

Based on the resident population number of Shanghai in 2017, the city still has an overall population carrying potential of 1293.30–7863.97 thousand persons and the appropriate population carrying potential of 4578.64 thousand persons. Furthermore, there are great spatial differences in the land population carrying potential. The population in the urban centre area exceeds the lower limit of the population carrying capacity estimation and the appropriate population carrying capacity, especially in the Hongkou and Huangpu District. In addition, the suburbs, especially the outer ones, have a larger population carrying potential.

Overall, although population growth may affect resources availability and eco-logical environment quality, resource scarcity and environmental capacity will also restrict population growth, especially land and water resources, and the environmental quality in large cities. Simultaneously, socio-economic and technological development and government behavior will enhance land use intensity, increase economic output, strengthen urban infrastructure construction and improve water, soil and air quality. In turn, these changes will affect the urban land population carrying capacity. Because of this, urban land carrying capacity is considered to be dynamic, while the method proposed in the paper may be used for continuous dynamic updating, thus providing a typical case reference for similar studies of other large cities and urban agglomerations.

Thank you for your suggestions and comments again.

Sincerely yours,

Hefeng Wang

Round 2

Reviewer 1 Report

The changes made in the current version of the article respond to the submitted requests; the authors have also well framed the meaning and importance of the suggestions. The quality of the research work is considered to be of a good standard.